# Culture matters: A systematic review of antioxidant potential of tree legumes in the semiarid region of Brazil and local processing techniques as a driver of bioaccessibility

**Michelle Cristine Medeiros Jacob**[1]*, **Juliana Kelly da Silva-Maia**[2], **Ulysses Paulino Albuquerque**[3], **Fillipe de Oliveira Pereira**[4]

1 Laboratório Horta Comunitária Nutrir, Nutrition Department, Federal University of Rio Grande do Norte, Natal, Brazil, 2 Nutrition Postgraduate Program, Nutrition Department, Center for Health Sciences, Federal University of Rio Grande do Norte, Natal, Brazil, 3 Laboratório de Ecologia e Evolução de Sistemas Socioecológicos, Departamento de Botânica, Universidade Federal de Pernambuco, Recife, Pernambuco, Brazil, 4 Biochemistry Laboratory, Academic Unit of Health, Education and Health Center, Federal University of Campina Grande, Cuité, Paraíba, Brazil

☯ These authors contributed equally to this work.
* michelle.jacob@ufrn.br

**Data Availability Statement:** All relevant data are within the manuscript and its Supporting Information files.

## Abstract

Ethnobotanical studies report that human populations from the Brazilian Caatinga biome use tree legumes (Fabaceae) with medicinal and food purposes. Our study provides a systematic review of the available published information concerning the antioxidant potential of *Hymenaea courbaril* L. (jatobá), *Libidibia ferrea* (Mart. Ex Tul.) L.P.Queiroz (jucá), and *Dioclea grandiflora* Mart. Ex Benth. (mucunã). Furthermore, in this paper, we infer the possible effects of local processing techniques applied to these plants on their antioxidant potential. In order to achieve these goals, we reviewed 52 articles, including studies from ethnobiology (n = 17), chemistry (n = 32), and food studies testing antioxidant activity (n = 17), excluding 14 repetitions. We found that these legume species can inhibit the formation of free radicals and this potential action varies among different parts of the plant. Probably, the presence of phenolic compounds such as phenolic acids and flavonoids, which are not uniformly distributed in the plants, explain their antioxidant activity. Local processing techniques (i.e., roasting, milling) affect the bioaccessibility of antioxidant components of tree legumes, inducing both positive and negative effects. However, studies about the antioxidant potential did not consider local processing techniques in their analyses. Our study highlights that culture is a fundamental driver of nutritional and pharmacological outcomes related to edible resources since it determines which parts of the plant people consume and how they prepare them. Hence, ignoring cultural variables in the analysis of antioxidant activity will produce inaccurate or wrong scientific conclusions.

**Funding:** The National Institutes of Science and Technology in Ethnobiology, Bioprospecting, and Nature Conservation, certified by CNPq, provided financial support from Facepe, the Foundation for Support to Science and Technology of the State of Pernambuco to UPA (Grant number: APQ-0562-2.01/17). The Brazilian Coordenação de Aperfeiçoamento de Pessoal de Nível financed the fee to publish this article (Finance Code 001). The funders had no role in study design, data collection and analysis, decision to publish, or preparation of the manuscript.

**Competing interests:** The authors have declared that no competing interests exist.

# Introduction

The Fabaceae family comprises species that have double relevance for human health. First, this is a group of plants of great pharmacological value, being widely used by human populations for medicinal purposes [1]. Second, legumes—as the dry seeds of edible Fabaceae are commonly called have been identified as central to the promotion of sustainable diets and food and nutritional security, given their nutritional potential for human and soil health [2,3]. Due to this dual function, medicinal and nutritional, many species of this family can be classified within the food-medicine continuum [4]. Among these plants, are the three analyzed in this article: *Hymenaea courbaril* L. (jatobá), *Libidibia ferrea* (Mart. ex Tul.) LPQueiroz (jucá), and *Dioclea grandiflora* Mart. ex Benth. (mucunã). In traditional communities in the Caatinga, the semiarid region in northeast of Brazil, these plants are consumed both for the prevention and treatment of specific diseases, as well as in daily meals, especially during dry periods [5,6]. Concerning Caatinga plants, many benefits to human health (e.g., anti-inflammatory, antioxidant) are related to the phenolic compounds present in these species [7]. Thus, our first objective with this review is to investigate the antioxidant potential of these three plants.

Furthermore, we argue that one of the factors that modulate the real antioxidant potential of these plants is culture, since before oral consumption these resources are processed based on recipes (e.g. *lambedor*—a medicinal syrup typical from Brazilian semiarid regions prepared with one or more herbs and sugar, all ingredients are cooked until a thick texture is obtained) and culinary techniques (e.g., *pilar—a traditional technique of mechanical processing that is used to mill grains or dry meat*). These procedures are culturally constructed and modified according to local knowledge, cultural heritage, and environmental factors. Scientifically, we can define processing as a series of operations (e.g., washing, macerating, roasting, grinding, fermenting) to convert unprocessed food into food for consumption, cooking, or storage [8]. We can extend this definition to encompass techniques applied to resources that humans ingest, regardless of their purpose (i.e., medicinal, food, recreational). Processing analysis is relevant to the problem presented in this article since phenolic compounds are relatively unstable. In this sense, changes inherent to processing (i.e., variation in temperature, pH, light, oxygen, enzymes, metal ions; or interaction with other constituents of the food matrix and other ingredients) are conditions that can change the bioaccessibility of phenolic compounds [9]. This variation can have either a negative effect generated by the instability of the compounds or positive due to the conversion to more active phenolics after the removal of glycosidic residues. Thus, our second objective with this paper is to evaluate the probable effects of local processing techniques applied to these plants on their antioxidant potential.

The processing analysis is critical in the case of Fabaceae, since legume seeds, especially when consumed as food, require processing before consumption to inactivate or remove antinutritional factors, which potentially can reduce the bioaccessibility of bioactive compounds. In an assessment of indigenous processing techniques for underutilized legumes, Sathya and Siddhuraju (2013) demonstrated that fermentation is the technique that best balances the trade-offs of inactivating antinutritional factors and preserving bioactive compounds [10]. Studying this same family, Vadivel and Biesalski (2012) present germination as an adequate treatment for velvet bean (*Mucuna pruriens* L. DC) because this technique is the one that best preserves bioactive compounds when compared with others [11]. These pieces of evidence reinforce the importance of the cultural bias approach in the assessment of nutritional and pharmacological outcomes, since the consumption of edible resources (i.e., consumed parts, processing, quantity) is strongly driven by human culture and behavior [12,13].

We address our goals by reviewing three types of studies: (1) ethnobiological, which informs us about the uses and processing techniques applied to these plants; (2) chemical,

which synthesizes the chemical profile of the evaluated species; and, finally, (3) antioxidant activity experiments, which inform us about the potential of bioactive compounds, considering experimental environments. We selected the plants for this study based on our earlier systematic review in which we prospected strategic plants capable of promoting food and nutrition security in the semiarid region [14]. One of the recommendations of this article is that the scientific community should focus its efforts on Fabaceae, which, due to their physiological and nutritional qualities, simultaneously promote human and environmental health, economic resilience, and sustainable agriculture. Therefore, starting from the original list of 65 plants [15], we focused on tree legumes identified in this review. We justify selecting tree species based on the argument that these plants have frequently been identified as strategic for semiarid regions due to their resilience to deal with water scarcity reinforced by climatic changes [16]. For safety reasons, we selected only those species with a record of consumption by human populations.

In short, in this study, by reviewing research from different areas, we summarize the current knowledge about the antioxidant potential of these plants and the likely effects of several processing techniques on this potential. During our investigation, we identified some gaps in research protocols from different areas that impeded our efforts to answer our questions properly. Thus, in this article, we also indicate these weaknesses in research protocols, with the expectation that by addressing them in future studies we will be able to design robust multidisciplinary research protocols.

## Material and methods

We conducted a systematic review based on the Preferred Reporting Items for Systematic Reviews and Meta-Analysis (PRISMA) Statement; see S1 Table. Our protocol for this review was not previously registered because our research does not directly analyze any health-related outcomes. In this review, we focused on tree legumes with consumption reported by human populations: *Hymenaea courbaril* L. (jatobá), *Libidibia ferrea* (Mart. ex Tul.) L.P. Queiroz (jucá), *Dioclea grandiflora* Mart. ex Benth. (mucunã). Once we analyzed the traditional knowledge associated with these plants, we registered this research with the Genetic Heritage Management Council (SisGen, in Portuguese) under number A1D4136.

### Selection criteria and search sources

We selected articles following these eligibility criteria: (i) original articles, published in English, Spanish, or Portuguese; (ii) papers focused on the study of antioxidant activity, and on the chemical and ethnobiological aspects of the plants we selected; finally, (iii) our investigation considered papers without time constraints. We excluded (i) repeated articles; (ii) review products; (iii) manuscripts without the identification of the plant and its registration in the herbarium [17]. We highlight that we excluded papers lacking proper identification of plants in order to reduce the risk of bias from individual studies.

During February 2021, FOP performed the search using four databases: Web of Science, Medline/PubMed (via the National Library of Medicine), Scopus, and Embase. The research consisted of applying the descriptors in each database. The following strategy guided the search: *("plant species")* AND *(antioxidant* OR *"oxidative stress"* OR *polyphenol)*. Basionym of taxa were not considered in the search.

### Study selection

With the assistance of the tool Rayyan, all records were organized and duplicates were deleted. Using the same tool, initially, titles and abstracts underwent a first screening, at which point

we excluded those that did not meet the selection criteria. Then, we proceeded to a full reading of potentially eligible texts.

## Data extraction

We extracted data from the selected articles into three spreadsheets designed to answer our research questions. Three authors were involved in the extraction (FOP, JKSM, MCMJ). First, we gathered the following information in ethnobiology studies: (i) part used; (ii) use (if medicinal or food), (iii) culinary processing technique; MCMJ verified the accuracy and scope. Second, in chemistry studies, we gathered the following information: (i) plant species; (ii) part used; (iii) main chemical classes (e.g., phenolic compounds) or bioactive compounds (e.g., dioclein); FOP was responsible for verifying the accuracy and scope. Finally, from the studies on antioxidant potential, we gathered: (i) article data (authors, year of publication); (ii) location of the study and collection of plant material; (iii) plant species; (iv) part used; (v) extraction conditions; (vi) antioxidant tests; (vii) main effects; (viii) conclusions; JKSM verified the accuracy and scope.

## Summary of results

Considering the heterogeneous nature of the studies, we produced different tables for each category (i.e., one each for ethnobiology, chemistry, and antioxidant potential studies). For each of the three tree legumes we present the data by parts used (e.g., leaves, roots, fruits).

# Results

## Study selection

The search in the databases led to the recovery of 726 studies (39 in the Web of Science, 22 in Medline/PubMed, 643 in Scopus, and 22 in Embase). After excluding 77 duplicates, we considered 649 articles as eligible for the next stage of selection, among them: 388 to *H. courbaril*; 191, *L. ferrea*, and 70, *D. grandiflora*. Based on titles and abstracts, we selected papers for a full reading. At this stage, we excluded those articles that evaluated activities unrelated to antioxidants (e.g., vasorelaxant activity), those that assessed isolated molecules instead of plants, research that did not address at least one of the three plants studied, and, finally, papers on research methods. In addition, we excluded five papers on *H. courbaril* because they did not adequately identify the species. We divided papers into the following categories: ethnobiology (n = 17), chemical profile (n = 32), and antioxidant activity (n = 17), see S2 Table. Considering that 14 of these papers are duplicated in more than one category, a total of 52 articles makes up this review. Fig 1 shows the study selection process and the related flowchart.

## Summary of results on uses of these plants by humans in ethnobiology studies

In this review we summarize several processing techniques used by human populations to prepare *H. courbaril*, *L. ferrea*, and *D. grandiflora* before oral intake (Fig 2).

We summarize the cultural uses of this plants, reported in the 17 studies reviewed in Table 1.

We highlight that 6 of the 17 studies that provide information on human uses of these plants were set in Brazilian biomes other than Caatinga, specifically Amazon, Cerrado, and Pantanal. Even if the plants we are studying are not native to these biomes, they may have been introduced into them through trade, exchange, or importation and now have their consumption incorporated by local communities. As our goal was to gather data that described different

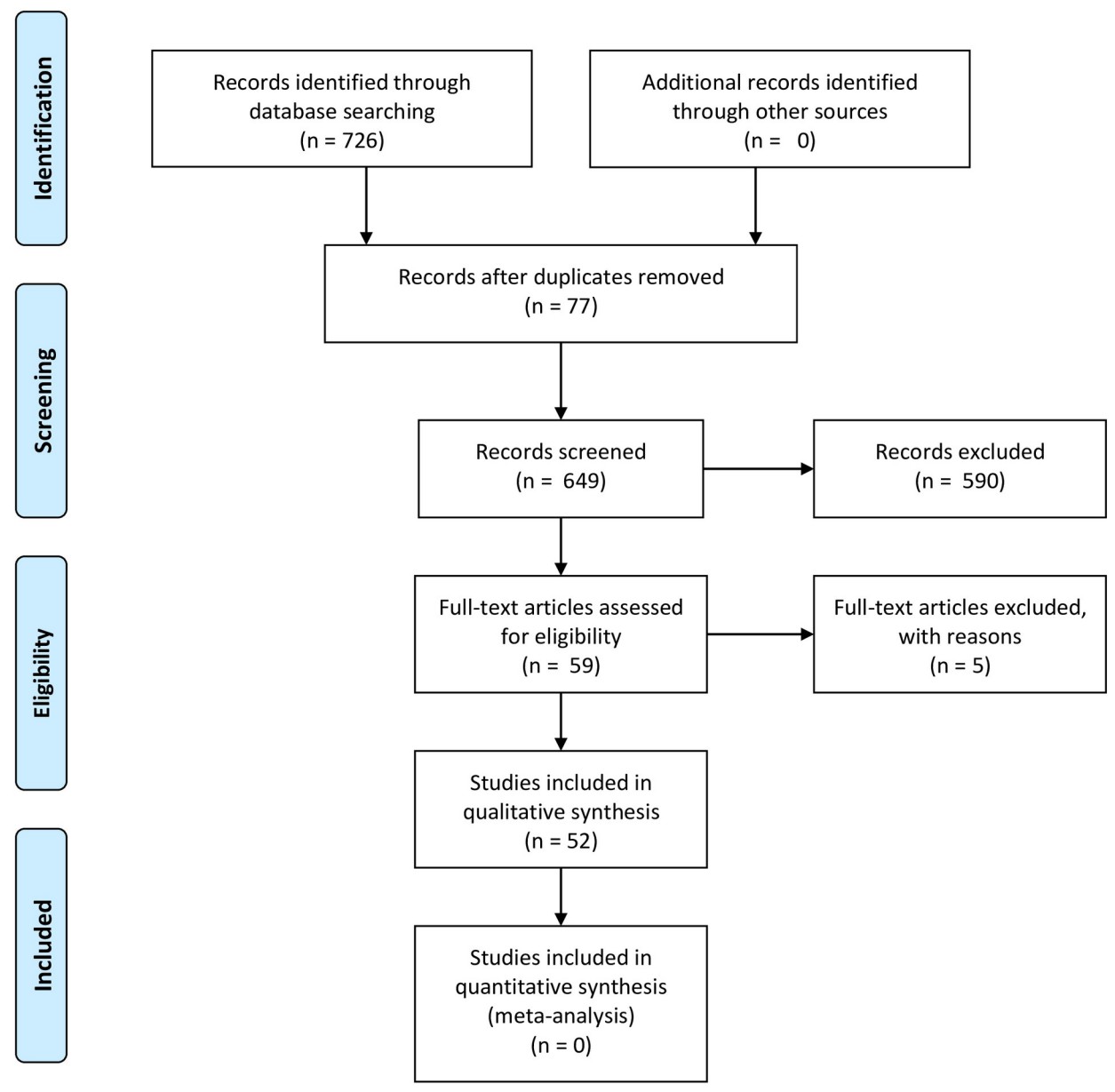

**Fig 1. Flowchart of the study selection process.**

forms of use, we included the information provided by these studies in our analysis. As represented in Table 1, we found more studies dedicated specifically to exploring medicinal uses (n = 14) than food uses (n = 3).

Considering our goal of evaluating the effect of local processing techniques on the antioxidant activity of *H. courbaril*, *L. ferrea*, and *D. grandiflora*, a strength of the ethnobotanical studies we reviewed is their concern with the proper identification of the plant and its registration in herbariums. However, their weakness, considering our purpose, is the lack of information about processing techniques, expressed in different forms in the studies: absence of data on how people process plants; a misconception of processing techniques, leading researchers to register products instead of the technique itself (e.g., tea instead of infusion or decoction or

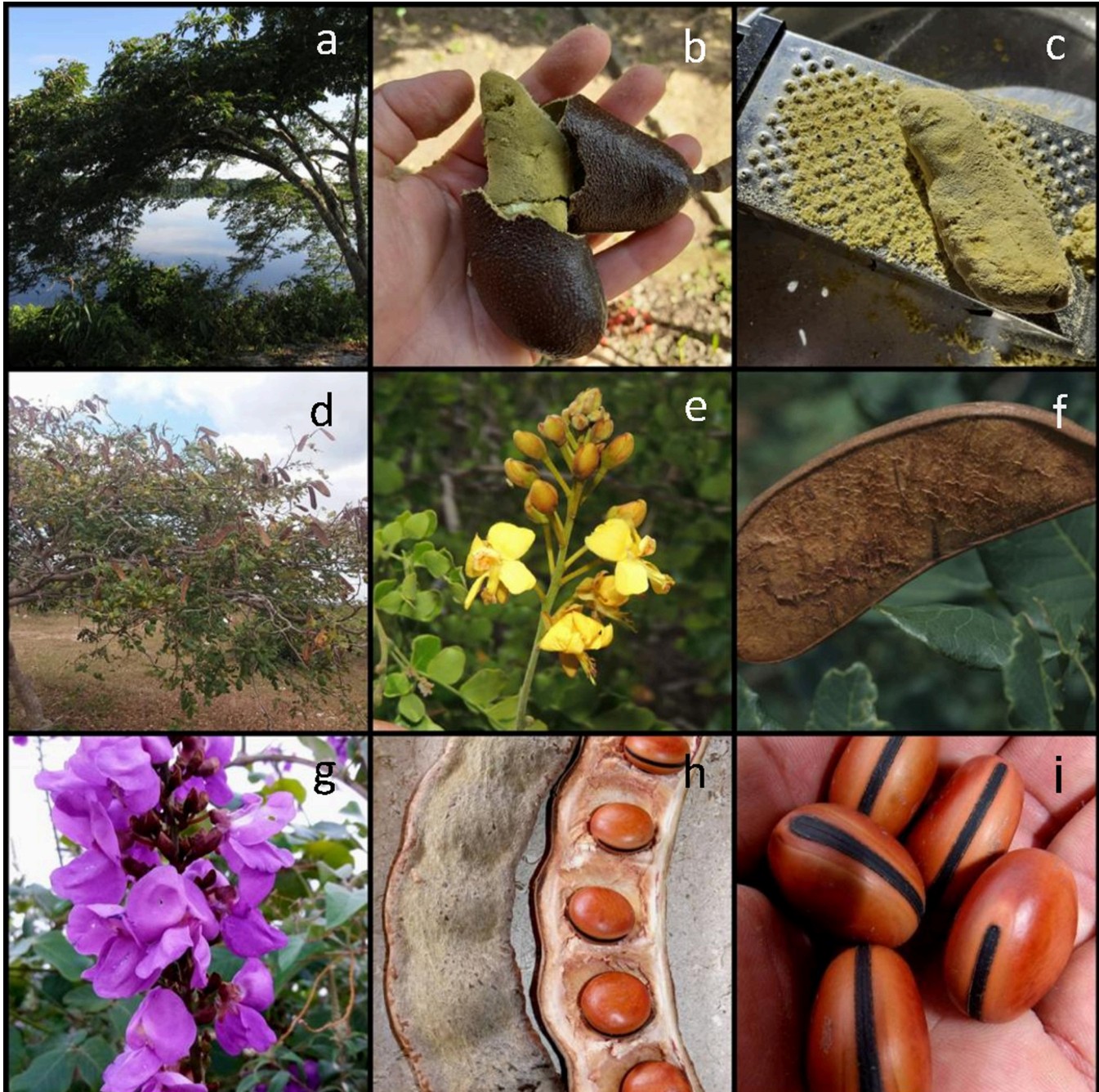

**Fig 2. Tree Legumes analyzed in our review.** (a) Tree, (b) whole fruit, and (c) extraction of pulp fruit of *Hymenaea courbaril*, by Gabriela Gonçalves. (d) Tree, (e) flower, and (f) whole fruit of *Libidibia ferrea*, by Rocicler Silva, Kew Science, and Rubes Queiroz, respectively. (g) Flower, (h) whole fruit, and (i) seeds of *Dioclea grandiflora*, by Gildásio Oliveira.

maceration; flour instead of roasting and milling); documentation problems, making inaccurate the correspondence between parts of the plant used and processing techniques (e.g., parts: bark, root; methods: tea, bottle, soaking) or imprecision in the register of the parts used. Once we are working with *Fabaceae* plants, it is worth highlighting that, considering the botanical definition, we understand that when authors use pods and fruits in the original paper, they are

**Table 1. Synthesis of processing techniques applied to tree legumes before oral intake.**

| Plant | | Medicinal use | Food use |
|---|---|---|---|
| Name | Part used | Processing technique | |
| *Hymenaea courbaril* L. | Fruit | Unclear [18]. Not specified [19] | In nature [20]. Not specified [1] |
| | Stem bark | Not specified [1]. Unclear [20–22] | Not found |
| | Bark | Tea (decoction), thick syrup [23]. Not specified [7,24];. Maceration [25]. Unclear [26] | Not found |
| | Seed | Unclear [20,21] | Not found |
| | Stem inner bark | Immersed in [20]. Not specified [1]. Unclear [22] | Not found |
| | Stalk | Decoction, alcohol [27] | Not found |
| | Resin | Not specified [24] | Not found |
| | Inner bark | Unclear [26] | Not found |
| *Libidibia ferrea* (Mart. ex Tul.) L.P.Queiroz | Fruit | Tea (decoction), tea (maceration) [28]. Bottled recipe, infusion, thick syrup, soaking in water, alcohol, natural [29] Unclear [25] | Flour [6] |
| | Leaf | Tea (decoction) [30] | Not found |
| | Root | Decoction [27] | Not found |
| | Seed | Tea [31]; Unclear [20,25]. Not specified [1] | Flour [6] |
| | Stem bark | Decoction [1]. Unclear [20]). Not specified [1] | Not found |
| | Bark | Not specified [7]. Unclear [25] | Not found |
| | Inner bark | Immersed in water [26] | Not found |
| *Dioclea grandiflora* Mart. ex Benth | Seed | Unclear [20]. Not specified [1] | Exhaustive dialysis [18]. Roasted, powdered, and washed nine times to be used as food during the droughts [19] |
| | Stem bark | Unclear [20]. Not specified [1] | Not found |

[a]All terms used in this table are consistent with the data presented by the authors in the original papers. When the processing method was not available in the paper, we assigned the label "not specified" in our table; cases in which the parts of the plant and the method were not correlated, we categorized as "unclear"; finally, cases for which we did not find any use report in the literature, we labeled "not found."

referring to the same plant part. However, a more specific register is needed to understand better the parts consumed (e.g., whole fruit, pulp fruit). Taking these weaknesses into account, we lack accurate data to reach the second goal we proposed for this review.

Specifically, on food use, we know that local communities: use the fruit of *H. courbaril* without processing (i.e., in nature); process the fruit and the seed of *L. ferrea* into a flour, without specification of the method; and, finally, we know that they process the seed of *D. grandiflora*, into flour using exhaustive dialysis (in the local system, *lavar em várias águas*), a method that combines thermal (roasting) and mechanical techniques (milling), with several washing cycles. Therefore, the primary food use of these tree legumes seems to be in the form of flour, the final product in the three cases. The flour, probably, is an intermediate ingredient for recipes, such as *couscous*, cookies, cakes, breads, etc. In these final dishes, the processing involves extra thermal treatment (i.e., boiling, baking) and the addition of external ingredients (e.g., sugar, salt).

Concerning medicinal use, excluding data labelled unclear or not specified, we have the following results. For *H. courbaril*, we know that people process the bark by using heat (decoction), immersion (maceration), and by cooking it with other ingredients (such as sugar) to produce a thick syrup; the steam inner bark is also processed through immersion; finally, the stalk can be treated with either heat or alcohol. For *L. ferrea* we have a variety of parts used by people, mainly inner bark (immersion) and also leaf, root, and steam bark (decoction). Leaves are also prepared as tea, but without specification of the processing method. Unfortunately, the processing techniques used to prepare the steam bark of *D. grandiflora* are labelled unclear or not specified. The studies do not make clear, but by its cellulose content, we believe that

when bark or stalks are processed with alcohol or water (decoction), the liquid product is the part that people consume. Therefore, thermal techniques seem to be used widely to process plants with medicinal purposes.

As for the parts of the plants that had medicinal uses in intersection with food use, we highlight the thermal and mechanical processing, in addition to the occasional presence of alcoholic extraction. We emphasize the following processes: *H. courbaril* fruit, heat treatment (decoction, infusion); *L. ferrea* fruit, heat treatment (decoction and infusion), alcohol treatment, processing that possibly involves previous mechanical treatment (e.g., maceration), and techniques that involve more than one type of processing simultaneously (e.g., bottled recipe); *L. ferrea* seeds, likely heat treatment (tea).

For all plants, processing techniques labelled unclear include licking, tincture, juice, and immersed in water (These terms are consistent with the data presented by the authors in the original papers). In addition to the nomenclature problem—focusing on products instead of techniques—we also perceive an enormous eclecticism in naming processing methods. We imagine that some nomenclatures are equivalent in the context used (e.g., immersed in water equals maceration), but we cannot affirm that, since the original research does not specify. Besides, there are references to vernacular names (e.g., bottled recipe or *garrafada*) without details of the processing. These imprecisions make it difficult to infer impacts on the nutritional profile.

## Summary of results on chemical profile

Researchers leading the studies reviewed in this paper used several methods to analyze phenolic compounds in tree legumes occurring in Caatinga. We found that some studies contained a preliminary phytochemical analysis and others presented a structural characterization of components. Preliminary phytochemical analyses usually show only the presence of classes of compounds or chemical groups in the plant extract and provide general information such as phenolic compounds, flavonoids, tannins, saponins, and terpenoids. The structural elucidation of compounds, on the other hand, provides more specific information.

We present the available data on the chemical profile *H. courbaril*, *L. ferrea*, and *D. grandiflora* in Table 2. Phenolic compounds including phenolic acids and flavonoids were the most detected groups of compounds in plants. Other groups of metabolites are also present in plants such as alkaloids, tannins, saponins, cinnamic derivatives, and terpenoids.

Particularly, the fruit pulp of *H. courbaril* contains sucrose, linolenic acid, fibrous material, and a low content of phenolic compounds. In contrast, the fruit peels are rich in phenolic compounds, flavonoids, and terpenoids. Other parts of *H. courbaril* such as bark, leaves, and seeds also have phenolic compounds, flavonoids, tannins, saponins, and terpenoids. The stem bark should be highlighted because astilbin and anthocyanins group flavonoids were detected in their composition. Ellagic acid, a well-known phenolic acid, was the prominent component in *L. ferrea* seeds. Ellagic acid, gallic acid, and quinic acid are also found in bark, leaves, and fruits of the same plant. Moreover, flavonoids, tannins, and other phenolic acids have also been detected in several plant parts such as leaf, bark, branch, and fruit. The seeds of *D. grandiflora* have not been analyzed chemically. However, phytochemical studies show its ability to produce flavonoids such as agrandol, paraibanol, diosalol, b-amyrin, 5,7,2',5'-tetrahydroxy-6-methoxy-8-prenylflavanone, dioclein, dioclenol, and dioflorin, all of them found in the root bark; as well as floranol detected in its roots.

## Summary of results on antioxidant activity

The researchers applied different methods to investigate the antioxidant activity of Caatinga legumes. The most common extraction conditions were aqueous, methanolic, ethanolic, and

**Table 2. Chemical profile of food tree legumes occurring in Caatinga.**

| Plants | Part used | Main chemical classes/bioactive compounds |
|---|---|---|
| *Hymenaea courbaril* L. | Bark | Astilbin (flavonoid), anthocyanins (flavonoid), tannins, saponins, and terpenoids [32] |
| | Fruit peel | Phenolic compounds and flavonoids [33,34], essential oils [35], Crotomachlin (diterpenoid), labd-13E-en-8-ol-15-oic acid (diterpenoid), labdanolic acid (diterpenoid), (13E)-labda 7, 13 dien-15-oic acid (diterpenoid), labd-8 (17) (diterpenoid), 13E- dien-15-oic acid (diterpenoid), methyl ester of labd-13E-en-8-ol-15-oic acid (diterpenoid), and spathulenol (sesquiterpene) [36] |
| | Fruit pulp | Sucrose, linolenic acid, fibrous material [36], and low phenolic compounds content [33] |
| | Leaves | Phenolic compounds [33] |
| | Seeds | Phenolic compounds [33,37–39] |
| *Libidibia ferrea* (Mart. ex Tul.) L. P. Queiroz | Bark | Phenolic compounds [39,40], tannin, flavonoid, and coumarin [40], gallic acid (phenolic acid), quinic acid (phenolic acid), kaempferol (flavonoid) [41,42], ellagic acid (phenolic acid) [43], tannins [44], catequin (flavonoid) [41], and epicatechin (flavonoid) [45] |
| | Branch | Phenolic compounds, flavonoids, triterpenes, and saponins [46] |
| | Fruits | Phenolic compounds [42,46–51], flavonoids, triterpenes, saponins [46], tannins, catechin (tannin), epicatechin (tannin) [42,44,45,52], galloylquinic acid (tannin), galloyl-HHDP-hex (tannin), brevifolin carboxylic acid (tannin), valoneic acid dilactone (tannin), ellagic acid derivative (ellagic acid hex-) (phenolic acid), and dihydroisovaltrate [49], ellagic acid (phenolic acid), gallic acid (phenolic acid) [47,49,53–56] |
| | Leaves | Phenolic compounds [46,57], flavonoids, terpenes, saponins [46,58], alkaloids, cinnamic derivatives, heptacosane (alkane) [58], tannins [57,58], quercetin (flavonoid) [59], ellagic acid (phenolic acid), gallic acid (phenolic acid) [51,60] |
| | Seeds | Ellagic acid (phenolic acid) [43] |
| *Dioclea grandiflora* Mart. ex Benth | Root bark | Dioclein (flavonoid) [61,62], Dioclenol (flavonoid) [63], Dioflorin (flavonoid) [64], agrandol (flavonoid), paraibanol (flavonoid), diosalol (flavonoid), b-amyrin (flavonoid), and 5,7,2',5'-tetrahydroxy-6-methoxy-8-prenylflavanone (flavonoid) [65] |
| | Roots | Floranol (flavonoid) [62] |

hydroethanolic. Experiments included *in vitro* and *in vivo* tests. In order to analyze the radical scavenging action from samples, among papers we reviewed authors commonly performed DPPH, 2,2-difenil-1-picrilhidazil radical scavenging activity tests. They also performed ABTS, 2,2'-azino-bis (3-ethylbenzthiazoline-6-sulphonate) radical scavenging activity tests. Considering *in vitro* tests, an inhibitory concentration (IC50) of 50% compared to standard samples indicates antioxidant activity. Finally, regarding *in vivo* tests, the analysis of lipid peroxidation involved TBARS, thiobarbituric acid reactive substances test, as well as activity and expression of antioxidant enzymes.

In Table 3, we present antioxidant data on tree legumes extracted from the selected studies. A total of 17 papers reported the antioxidant potential of *L. ferrea* ($n = 10$), followed by *H. courbaril* ($n = 6$), and *D. grandiflora* ($n = 1$).

Researchers report that bioactive compounds in the fruit of *H. courbaril* inhibit lipid peroxidation and reduce the damage caused by reactive oxygen species [36,68]. Authors also found the lowest $IC_{50}$ value by analyzing isolated seeds compared to other parts of the plant, finding the highest antioxidant activity in seeds according to DPPH and FRAP results [68]. By running DPPH tests, other authors found that either ethanolic, hydroethanolic, and methanolic extracts from *H. courbaril* seeds had antioxidant potential similar to standard samples of ascorbic acid,

**Table 3. Antioxidant activity of tree legumes occurring in Caatinga.**

| Plant | Parts used | Extraction conditions | Antioxidants tests | Main Effects | Conclusions | Reference | Setting |
|---|---|---|---|---|---|---|---|
| *Hymenaea courbaril L.* | Fruits (edible powder and seed pod) | Isolated compounds: Terpenes | Lipid peroxidation inhibitory activity | Inhibition of the lipid peroxidation: crotomachlin (46%), labd-13E-en-8-ol-15-oic acidand (48%), (13E)-labda 7, 13 dien-15-oic acid (75%) | The compounds isolated from this fruit inhibited lipid peroxidation, alleviating the damage caused by reactive oxygen species | [66] | Kingston, Jamaica |
| *Hymenaea courbaril L.* | Seed | Ethanolic extracts: Powdered material (500 g) was submitted to extraction with 99% ethanol (1.5 L) at room temperature (25–27˚C) for 3 days, the process was repeated twice. The extract was concentrated by in rotary evaporator | DPPH | Similiar IC50 compared to the starndard. Extract—247.95 µg/mL; Standard Vitamin C: 260.27µg/mL | The extract showed antioxidant activity by inhibiting the oxidation of DPPH radical | [67] | Araripe National Forest, CE —Brazil |
| *Hymenaea courbaril L.* | Seed | Hydroethanolic extract: maceration with 70% ethanol-water for 24h (100 mg/mL), at room temperature, in the dark with constant stirring (twice). | DPPH | Extract's antioxidant capacity did not significantly differ from gallic acid standard. Seed extract: 78.94 ±0.46% Gallic acid 79.98±1.60% | 70% ethanolic extract showed antioxidant activity | [38] | Assis, AC— Brazil |
| *Hymenaea courbaril L.* | Seed | Maceration with ethyl acetate (3x) and ethanol (4x) followed by fractionation of the extract. Products: Methanol fraction, hexane fraction, and ethyl acetate extract | DPPH | Antioxidant capacity of methanol fraction was like standards. Methanol fraction: 96.10% Ethyl acetate extract: not determined Ascorbic acid: 93.57% Rutin 96.52% | Methanol fraction showed antioxidant activity and protected the liver and kidneys against oxidative stress and lesions induced by acetaminophen. | [37] | Nova Canaã do Norte, MT—Brazil |
| *Hymenaea courbaril L.* | Leaves, fruit rinds, pulp, and seed | Ethanolic and methanolic extracts: Maceration with metanol or ethanol solution (70% ethanol-30% water) for 24 h in (100 mg/mL), at room temperature in the dark with constant stirring. The process was repeated twice. The 70% ethanol extract from the seeds was fractionated in ethyl acetate/methanol (EAM 70:30 v/v), ethyl acetate/methanol (EAM 50:50 v/v), ethyl acetate/methanol (EAM 30:70 v/v) and methanol (MT) | DPPH / FRAP / ORAC | The 70% ethanol extract from seeds produced the lowest IC50 value (149.45 µg mL–1). Lower IC50 values indicate higher antioxidant activity, which was corroborated by FRAP results DPPH: Ethanolic extract: Leaves (415.80 µg/mL), Fruit rinds (428.10 µg/mL), seeds (149.45 µg/mL); Methanolic extract: Leaves (392.05 µg/mL), Fruit rinds (395.44 µg/mL), seeds (179.43 µg/mL), EAM 70:30 (72.21±1.08%), EAM 50:50 (5.65.±2.62%), EAM 30:70 (7.24±0.76%), MT (4.90 ±3.02%), Acid gallic (43.82). FRAP: Ethanolic extract (µMTE.g-1 of extract): Leaves (632.64 ± 08.20), Fruit rind (1274.42 ± 59.42), seeds (3073.51 ±66.73); Methanolic extract (µMTE.g-1 of extract): Leaves (1112.63 ±53.24), Fruit rind (614.31±21.72), seeds (2797.90 ±28.83), EAM 70:30 (3029.97 ±09.78), EAM 50:50 (118.98. ±50.77), EAM 30:70 (122.67 ±40.74), MT (277.76±22.85), Trolox results not informed. ORAC: Ethanolic extract (Trolox equivalent): Leaves (0.34), Fruit rind (0.28), seeds (0.25); Methanolic extract (Trolox equivalent): Leaves (0.34), Fruit rind (0.16), seeds (0.12) | The seed ethanolic extract showed the strongest antioxidant activity, but all extracts demonstrated antioxidant activity and high phenolic compound content, except the pulp extract. | [68] | Assis, AC— Brazil |
| *Hymenaea courbaril L.* | Bark | Ethanol extract: maceration at room temperature for 7 days and fractions (dichloromethane, dichloromethane:ethyl acetate, ethyl acetate, and methanol) | DPPH | IC50 results (µg/mL)—Ethanol extract:3.07±0.18; Dichloromethane fraction: 66.3± 6.9; Dichloromethane:ethyl acetate fraction 34.0±0.24; Ethyl acetate fraction: 5.05±1.5; Methanol fraction: 5.12±0.73; Standard (Trolox) 2.6±0.23. | The extract and its fractions showed antioxidant activity by inhibiting the oxidation of DPPH radical. The strongest activities were ethanol extract > methanol fraction > ethyl acetate fraction | [69] | Crato, CE— Brazil. |

(*Continued*)

**Table 3.** (Continued)

| Plant | Parts used | Extraction conditions | Antioxidants tests | Main Effects | Conclusions | Reference | Setting |
|---|---|---|---|---|---|---|---|
| *Libidibia ferrea* (Mart. ex Tul.) L. P. Queiroz | Fruits | Hydroethanolic extract: cold maceration with 40% hydroalcoholic solvent for 3 days | ABTS / DPPH / TAA/ Superoxide Radical (O2 -) Scavenging Activity / TBARS (*in vivo experiment*) | Lower IC50 than control (Trolox) in ABTS and DPPH tests. TAA results are close to control (ascorbic acid). Oral pretreatment decreased lipid peroxidation levels by 36.05 to 44.19% in Wistar rats with the absolute ethanol-induced gastric lesion. | Its antioxidant activity was one of the key mediators of gastroprotective effects observed in the experimental model of gastric lesions. | [70] | Barbalha, CE—Brazil |
| *Libidibia ferrea* (Mart. ex Tul.) L. P. Queiroz | Fruits | Solvents: n-hexane (HEX), chloroform (CLO), ethyl acetate (ACO) and alcohol 70% (AE). The solvent: material ratio was 2:1. Ultrasound-assisted extraction 30°C (± 3) and 30 min, except to ethanol (45 ± 3°C) | ABTS/ DPPH/ ORAC | The ethanol and ethyl acetate extracts both had antioxidant activity, the ehtanol extract showing greater potential. DPPH and ABTS assays showed that the AE and ACO extracts exhibited dose-dependent antioxidant activity, whereas the CLO and HEX extracts showed no such activity. AE showed highest antioxidant activity with the lowest IC50. The ORAC test confirmed the antioxidant capacity of the AE extract. | Aqueous ethanol extract can function as an exogenous antioxidant *in vitro*, and thus could potentially act against oxidative stress-related diseases and/or strengthen the health and well-being of an organism. The antioxidant action from the hydroalcoholic extract was associated with phenolic compounds presence. | [71] | Maraba, PA —Brazil |
| *Libidibia ferrea* (Mart. ex Tul.) L. P. Queiroz | Fruits | Aqueous extract: 7.5% plant material (w/V), boiling water as a solvent and 15 minutes of extraction. | ABTS/ DPPH / Superoxide anion radical scavenger activity/ β -carotene bleaching method/ Cell antioxidant activity (NIH-3T3 cells)/ TBARS (*in vivo* experiment) | Higher IC50 than standards (ascorbic acid and gallic acid) in ABTS, DPPH, and Superoxide assays. Higher IC50 in Antioxidant activity by β-carotene/linoleic acid system than standard (BHT). 40% inhibition of oxidation in the cell-base antioxidant assay at 50 ug/ml dose (inhibition rate from standard, quercetin, was 80%). Significant decrease in the serum lipid peroxidation (TBARS assay) in mice with acute hepatic injury induced by CCl4 and treated with100 and 200 mg/ml. | Hepatoprotective activity associated with antioxidant activity | [45] | Manaus, AM— Brazil, |
| *Libidibia ferrea* (Mart. ex Tul.) L. P. Queiroz | Fruits | Hydroethanolic extract: ethanol: water (7:3) by agitation of 300 rpm for 15 h | Total antioxidant capacity by phosphomolybdenum assay/ Reducing power assay / Superoxide radical scavenging assay/Hydrogen peroxide radical scavenging assay / Nitric oxide radical scavenging assay | Total antioxidant was 38.06% (±2.04) in relation to standard activity (ascorbic acid). The IC50 in Reducing power and Superoxide radical scavenging assays were 2.5x 4.7x higher the standard (gallic acid), but the IC50 concentration in Hydrogen peroxide radical scavenging and Nitric oxide radical scavenging assays were like standard (gallic acid). | The results obtained in this study demonstrate that fruits of L. ferrea exhibited antioxidant activity in the tested *in vitro* assays. | [50] | Parna do Catimbau, PE—Brazil, |
| *Libidibia ferrea* (Mart. ex Tul.) L. P. Queiroz | Fruits | Hydroethanolic extract: 10% (m/v) using the following as solvent: water or hydroalcoholic-20-80% ethanol by turbidysis. | Total glutathione content / MDA content | All extracts (aqueous and hydroethanolics) were able to significantly increase the total glutathione level and to reduce the MDA level in experimental animals with induced peritonites | The fruit showed important anti-inflammatory, antioxidant, and peripheral antinociceptive effects *in vivo*. The biological activity of the analyzed species is probably related to the presence of bioactive compounds. | [56] | Limoeiro, PE—Brazil. |
| *Libidibia ferrea* (Mart. ex Tul.) L. P. Queiroz | Leaves | Aqueous extract: 10% (w/v) by turbo extraction (four extractive cycles of 30 s, with 5 min of pause). Solvent was water. | Total glutathione content / MDA content—*in vivo* assays | All doses (100, 200 and 300mg/ kg) were able to significantly prevent the reduction of total glutathione levels in the arthritis experimental model. Significantly reduced MDA levels, being 60% decreased at doses of 200 and 300 mg/kg. | The observed anti-inflammatory potential can be related to bioactive compounds, including those with antioxidant action. | [72] | Recife, PE– Brazil |

(*Continued*)

**Table 3.** (Continued)

| Plant | Parts used | Extraction conditions | Antioxidants tests | Main Effects | Conclusions | Reference | Setting |
|---|---|---|---|---|---|---|---|
| *Libidibia ferrea* (Mart. ex Tul.) L. P. Queiroz | Leaves | Hydroethanolic extract: 70% aqueous ethanol under reflux | DPPH / *In vivo*: GPX / CAT / TBARS / GSH | Exhibited a remarkably rats with induced diabetes, the extracts in both concentration (250 and 500 mg/kg) significantly raised GPX, SOD and CAT enzyme activities and GSH level, and inhibited the formation of TBARS as compared to diabetic control group in a dose-dependent manner. Though such improvement in GSH and TBARS levels did not restore to the basal level of non-diabetic control group, GPX, SOD and CAT enzyme activities returned to normal basal values at the highest dose. | The antioxidant potential of L. ferrea leaves extract inhibiting the progression of oxidative stress in STZ-induced diabetic rats, and could be associated to other benetial results | [73] | Cairo, Egypt |
| *Libidibia ferrea* (Mart. ex Tul.) L. P. Queiroz | Leaves | Aqueous extract: 25 mg/mL in distilled water at 98±2˚C, and remained without further heating for 10 min. | DPPH / BLCA system | Different species were analysed and *L. ferrea* presented the highest antioxidant capacity among them, showing lowest IC50 and highest antioxidant activity index according to DPPH assay. It was presented highest antioxidant capacity in BLCA system | The *L. ferrea* infusion had the highest radical scavenging activity among 9 Brazilian herbs studied. It was comparable to antioxidant activity of the conventional Camellia specie | [51] | Belém, PA —Brazil |
| *Libidibia ferrea* (Mart. ex Tul.) L. P. Queiroz | Bark | Aqueous extract: 7.5% plant material (w/V), boiling water as a solvent and 15 minutes of extraction. | ABTS/ DPPH / Superoxide anion radical scavenger activity/ β -carotene bleaching method/ Cell antioxidant activity (NIH-3T3 cells)/ TBARS (*in vivo* experiment) | Higher IC50 than standards (ascorbic acid and gallic acid) in ABTS, DPPH, and Superoxide assays.Higher IC50 in Antioxidant activity by β-carotene/linoleic acid system than standard (BHT). 60% inhibition of oxidation in the cell-base antioxidant assay at 50 ug/ml dose (inhibition rate from standard, quercetin, was 80%). Significant decrease in the serum lipid peroxidation (TBARS assay) in mice with acute hepatic injury induced by CCl4 and treated with100 and 200 mg/ml. | Hepatoprotective activity associated with antioxidant activity | [45] | Manaus, AM—Brazil, |
| *Libidibia ferrea* (Mart. ex Tul.) L. P. Queiroz | Bark | Methanolic extract: powder material was suspended in methanol, homogenization of the material (60 <C, 2 h, filtered 2 x) | GSH / GPX / Nitrate/nitrite content / MDA content/ | It increased GSH levels by 65% and GPx activity by 72%; and reduced nitrate-nitrite levels by 73% and MDA content by 37% in mice with induced peritonitis | The present study demonstrates anti-inflammatory and antioxidant effects of the polysaccharide-rich extract. | [39] | Quixadá, CE—Brazil |
| *Libidibia ferrea* (Mart. ex Tul.) L. P. Queiroz | Bark | Hydroethanolic extracts: maceration (48 h) at a proportion of 1:10 (m/v) with 80% ethanol, repeated three times | DPPH | IC50 was twice the standard (ascorbic acid) (27.53±0.54 and 14.78 ± 1.40 μg/mL), but it did not statistically differ from *M. urundeuva* (Aroeira) which showed the strong antioxidant activity (IC50 = 16.46 ± 0.41 μg/ml) | *L. ferrea* demonstrated intermediary antioxidant capacity and ultraviolet type B sun protection factor among 15 species studies. | [40] | Altinho, PE- Brazil |

*(Continued)*

**Table 3.** (*Continued*)

| Plant | Parts used | Extraction conditions | Antioxidants tests | Main Effects | Conclusions | Reference | Setting |
|---|---|---|---|---|---|---|---|
| *Dioclea grandiflora* Mart. ex Benth | Roots | Isolated compound—Floranol (3,5,7,20-tetrahydroxy-6-methoxy-8-prenylflavanon) | Resistance of isolated-LDL to oxidation by cooper / Interactions with Cu(II) and Fe (III) by combination of UV–visible (UV–Vis), mass spectrometries, and eletrochemical studies | Floranol inhibited the LDL oxidation, in a dose-dependent manner. Copper and iron reduction is less favorable in the presence of floranol. | Hypothesis for the floranol antioxidant activity: floranol will sequester Cu(II) and/or Fe(III) preventing their reduction, thus preventing their effect on LDL oxidation. It may have an important role in the inhibition of lipid peroxidation, a property which could be beneficial in reducing atherosclerosis. | [74] | João Pessoa, PB—Brazil |

[a]2,2'-azino-bis(3-ethylbenzthiazoline-6-sulphonate) Radical Scavenging Activity (ABTS); 2,2-difenil-1-picrilhidazil Radical Scavenging Activity (DPPH); 50% inhibitory concentration (IC50); Catalase (CAT); Ferric reducing antioxidant power activity (FRAP); Glutathione peroxidase activity (GPX); Malondialdehyde (MDA) content; Oxygen Radical Absorbance Capacity test (ORAC); Reduced glutathione (GSH); Superoxide dismutase activity (SOD); Thiobarbituric acid reactive substances test (TBARS);Total antioxidant activity (TAA); β-carotene/linoleic acid (BCLA) system.

gallic acid, and rutin [75–77]. One study evaluating the bark of this species indicated the potential of this part to inhibit the DPPH radical close to the Trolox standard [69]. In the same study, the ethanol sample was the most potent, followed by methanol and ethyl acetate. Unfortunately, even though we know that there are phenolic compounds in *H. courbaril* leaves [68], to date, no studies have tested their antioxidant activity.

Researchers analyzing *L. ferrea* antioxidant activity tested several plant parts, including the fruit (pods), bark, and leaves. In the description of the plant parts, the authors used either the term *fruit* or the term *pods*, which are botanical synonyms. Although, as previously mentioned, we argue these terms are insufficient to inform the parts consumed. For example, whole fruit or fruit pulp would be better descriptions. In the following paragraph, we show a clear case in which a general description hampers the interpretation of data.

Prazeres et al. (2019) evaluated hydroethanolic extracts from pods and found a lower $IC_{50}$ than in the control sample, both in the DPPH and in the ABTS tests [70]. Additionally, oral treatment with pod-extract reduced lipid peroxidation in rats with gastric lesions. Another study showed that ethanol was the most efficient solvent to remove and preserve antioxidant compounds from *L. ferrea* pods. Moreover, the ethanol extract had higher antiradical action than ethyl acetate by DPPH and ABTS results [71]. The same study found that extracts with n-hexane or chloroform did not have antioxidant activity, but ethanol extracts had intense antioxidant activity, according to ORAC results. On the other hand, two experiments using a variety of tests (DPPH, ABTS, and superoxide assay) detected higher $IC_{50}$ in *L. ferrea* fruit than in standard samples [45,78]. It is useful to recall that the stronger the antioxidant power of the sample, the lower the concentration to promote 50% of radical inhibition ($IC_{50}$). Barros et al. (2014) analysed the fruit without seed [45], which can explain the lower antioxidant potential compared to the findings of Prazeres et al. (2019) [70]. In other analyses, the $IC_{50}$ was similar to the standard of gallic acid [50]. Furthermore, fruit extract promoted significant serum lipid peroxidation in mice with acute hepatic injury at doses of 100 and 200 mg/mL [45]. Another experimental assay Falcão et al. (2019a) showed that aqueous and ethanolic extracts boosted the antioxidant defense in mice with peritonitis, evidenced by the increased level of total glutathione and decreased MDA level [56]. Therefore, fruit with (whole pod) or without seed showed antioxidant activity, but in different degrees. Probably the seeds can explain part of this high antioxidant effect, but we are not sure about that because the description available in scientific reports is not sufficient to infer if the pods or fruit were used as a whole.

Still on *L. ferrea*, its bark exhibited higher $IC_{50}$ than standards in tests *in vitro* based on radical scavenging [40,45], but showed improvements in antioxidant defenses during experiments *in vivo*. The treatment with methanolic extract in mice with induced peritonitis increased GSH levels and GPx activity, besides reducing nitrate-nitrite levels and MDA content [39]. Additionally, the bark aqueous extract used in mice with acute hepatic injury reduced the serum lipid peroxidation [45].

Three studies present tests of the antioxidant activity from *L. ferrea* leaves. Aqueous extract in an arthritis experimental model reduced MDA levels and preserved the total glutathione levels [72]. Another assay concluded that hydroethanolic extract improved antioxidant status in diabetic mice [73]. The highest dose (500 mg/kg) recovered the GPx, SOD, and CAT enzyme activities back to the typical basal values. Besides, all doses led to lower TBARS results and increased the total GSH level. Finally, Port's et al. (2013) investigated leaves from nine Brazilian plants to test their antioxidant activity [51]. The authors concluded that *L. ferrea* leaves had the strongest potential among them. We highlight that the authors compared the antiradical potential of an infusion of *L. ferrea* leaves with another group of plants broadly recognized as potent antioxidants agents, which are *Camellia* spp (Theaceae).

Finally, the antioxidant activity from *D. grandiflora* is understudied. In our sample, only one study evaluates the antiradical plant action [74]. Furthermore, the analysis comprises only the root, which, according to ethnobotanical reports, humans usually do not consume. The analysis of Botelho et al. (2007) shows that floranol—the isolated compound from *D. grandiflora* roots—mitigates LDL oxidation in a dose-dependent manner [74]. The authors attributed the mechanism of action to the floranol ability to sequester Cu(II) and Fe(III), preventing their reduction and, consequently, avoiding LDL oxidation.

## Discussion

Theoretically, tree legume species included in this review demonstrated their ability to prevent the formation of free radicals. According to literature, there are differences in the potential antioxidant action among the evaluated plant parts. In addition, ethnobiological data indicate that the different methods applied to process these tree legumes (i.e., thermal, mechanical, washing) can change the bioaccessibility of bioactive compounds.

Concerning our first goal, we summarize here the antioxidant potential of tree legumes. *H. courbaril* seeds, bark, and fruit peel demonstrated antiradical action. *L. ferrea* bark, pods, and leaves also exhibited antioxidant capacity. While data show that people use *L. ferrea* seeds to obtain tea or flour, we cannot infer the antioxidant action of these foods and remedies because the authors that analyzed pods did not detail whether or not seeds were kept in them. *D. grandiflora* roots also demonstrated antioxidant properties. However, we do not know if these properties are preserved because there is no sufficient register in the literature of how people process them before intake.

Regarding our second objective, we conclude that culture, through processing techniques, modifies the potential antioxidant activity of compounds in these legumes. People prepare medicinal and food products from recipes that, in turn, embed ingredients and processing techniques from a specific cultural context [14]. Therefore, recognizing local culture is essential to understand how processing influences the antioxidant activity of plants consumed. In this sense, it is necessary to highlight that the chemical profile and antioxidant activity shown in this review for the tree species could be over or underestimated because most studies used extracts from raw materials ignoring cultural variables. In the following paragraphs, we infer some of the probable effects of local processing techniques in the stability of the bioactive compounds from studied species.

Domestic processing with the potential to affect the content and activity of phenolic compounds includes, for example, thermal methods (e.g., boiling, frying, steaming, cooking, braising, roasting) and mechanical techniques (e.g., peeling, trimming, cutting, mixing, milling) [79]. Unfortunately, there are no specific studies that assess the impact of processing techniques on the antioxidant compounds of *H. courbaril*, *L. ferrea*, and *D. grandiflora*. Therefore, to infer these probable influences, we present in this discussion some remarks from studies that report results of tests that included processing of plants with classes of phytochemicals similar to our legumes.

Taking our data into account, the primary product of tree legumes seems to be flour, processed from the seeds of *H. courbaril*, *L. ferrea*, and *D. grandiflora*. The flour preparation, in general, involves the combination of thermal, mechanical, and washing techniques. After that, flour can be an intermediate ingredient of new recipes (e.g., cookies, cakes) that will require new heat cycles. The medicinal use of these species also entails techniques with exposure to high temperatures, such as infusion and, especially, decoction. Thermal processing can have both beneficial and disadvantageous effects on edible resources [80]. The positive effects include: inactivation of harmful constituents and improvement of a range of factors including digestibility, bioavailability of nutrients, taste, and texture. The downside of thermal processing can involve: losses of certain nutrients, formation of toxic compounds, and production of compounds that create adverse characteristics in flavor, texture, and color.

Research shows how thermal methods modify the antioxidant potential of plants. Miglio et al. (2008), studying the impact of domestic cooking treatments on phenolic acids in carrot, zucchini, and broccoli, concluded that boiling, steaming, and frying decreased the total phenolic compounds in these vegetables [81]. Boiling had the most detrimental effect on carrot polyphenols, while steaming and frying had a less negative impact on total phenolics. Zhang et al. (2010) state that hot-air drying can cause phenolic compound damage [82]. In their assay, roasting the flour of buckwheat seeds at 120˚C to 160˚C for 20 to 30 minutes decreased the flavonoid content. Similarly, in the experiment of Lang et al. (2019) with black rice, an increment in drying temperature from 20˚C to 100˚C reduced the total content of free phenolics, free flavonoids, and anthocyanins [83]. In all these cases, thermal treatment promotes the instability of molecules leading to epimerization and breakdown of phenolic compounds [84,85].

In another direction, the following studies show that the content and the functionality of some phenolic compounds might also increase during heating. Dewanto et al. (2002), for example, reported that thermal processing at 100˚C to 121˚C for 10 to 50 minutes elevated the phenolic content and the total antioxidant activity of sweet corn [86]. Ferracane et al. (2008) also observed a significant increase in total caffeoylquinic acids in artichoke after boiling, steaming, and frying [87]. Another assay demonstrated that boiling and wet-cooking raised the concentrations of lipophilic antioxidants in whole grain rice [88]. An experiment with beans found that thermal processing at 100˚C caused a significant increase in the availability of total phenolic and antioxidant activity in both navy and pinto beans [89]. The mechanisms involved in raising antioxidant activity using heat are (1) formation of new compounds with antioxidant activity, such as Maillard reaction products, (2) biophysical changes that improve the extractability of phenolic compounds from the food matrix, and (3) conversion of antioxidant compounds into more active forms [85,90].

So far, it is clear that the effect of thermal processing on antioxidant molecules exists, but it varies in its direction. Moreover, this variation occurs considering the same compound in different food matrices. An example of this phenomenon occurs with one specific phenolic compound, ellagic acid (EA), one of the antioxidant molecules identified in *L. ferrea*seeds. We can find EA in two forms: free EA and EA bound with sugar molecules. EA is a highly thermostable molecule with a melting point of 450˚C and a boiling point of 796.5˚C, and stable in gastric

conditions [91]. Zafrilla et al. (2001), studying red raspberry, observed that jam cooking released two times more EA from the sample [92]. The high level of free EA and the lower gly-cosylated form suggest increased bioavailability, since the absorption of polyphenols depends on removing the sugar moiety [93]. Contrarily, Häkkinen et al. (2000) found 20% lower EA levels after processing berries into jam [94]. Therefore, processing has both positive and nega-tive effects on the activity of the same phenolic acid, depending on the plant. For this reason, the specific effects on EA remain inconclusive in the case of *L. ferrea*.

In the case of Fabaceae plants, thermal processing is necessary because it inactivates the antinutrients, contributes to nutrient bioavailability, and enhances flavor. To balance the dis-advantages of heat, a preliminary step—soaking—reduces the cooking time and promotes the loss of antinutritional factors by lixiviation [10]. Discarding the soaking water before cooking may eliminate phytates, phytic acid, and tannins, but it can also decrease the total phenolic compounds and antioxidant activity [95]. The loss of phenolic compounds is more significant when legumes (i.e., peas, chickpeas, fava beans, and kidney beans) are soaked than when they are not [96]. The effect is proportional to the length of time of soaking. Xu and Chang (2008) observed that soaking in water up to 16 hours decreased the total phenolic content and antiox-idant activity in green peas, yellow peas, chickpeas, and lentils [97]. This is what Sathya and Siddhuraju (2013) found in a study testing the effect of processing techniques in legume seeds of *Acacia auriculiformis* and *Parkia roxburghii* [10]. In the case of A. *auriculiformis*, for exam-ple, they observed a reduction of total phenolic content after soaking and thermal processing. According to the authors, the lixiviation of water soluble compounds is a factor that explains this loss.

In addition to thermal and soaking treatments, mechanical processing also affects the con-tent, availability, and activity of phenolic compounds. As antioxidant compounds are more preeminent in the external layers of plants, peeling results in reduced nutrient content [79]. The outer layer of grains and cereals, such as wheat, barley, and oat, has an especially high pro-portion of phenolic compounds [98]. Therefore, when people discard some parts of edible resources during the cleaning stage, they also waste phenolic compounds. Considering our data, this situation occurs when people consume the pulp, discarding the bark of *H. coubaril*. When the whole plant is processed, milling affects the distribution of phenolic compounds, making it more [98]. As previously stated, flour is the main product of processing legumes included in this review. A viable alternative to preserve bioactive compounds, in this case, is the production of whole flour, thus keeping the portions richer in phenolic compounds. It is also possible to use other parts of plants (e.g., bark, leaves, roots) when producing the flour. Milling can also improve the availability and activity of antioxidant compounds because it breaks the cells, affecting the plant matrix [99].

Literature has shown that fermentation improves the bioaccessibility of bioactive com-pounds. For this reason, we suggest that experimental food studies in the future start by testing the impact of fermentation on the antioxidant profile of the legume seeds studied here. Fer-mentation technique has been used through millennia to reduce the activity of antinutrients while preserving the bioaccessibility of other essential nutrients, as in the case of tofu processed from soybeans. Katz (1990) argues that by fermenting soybeans, the isoflavones can be better digested because microorganisms break down glycosylated isoflavones into aglycones [100]. Considering other plants, the assay of Verotta et al. (2018) showed that fermentation of pome-granate results in a sustainable source of ellagic acid, with intense antioxidant action [101]. Ifie and Marshall (2018) also argue that rice and wheat fermentation can increase phenolic acid levels [93]. Another study showed that a forty day fermentation of the seeds of *Pangium edule* Reinw, a tropical non-edible plant from Southeast Asia, results in edible products, such as the local spices *keluwak* and *kecap pangi* [102]. The explanation is that fermentation increases the

enzymatic activity of the β-glucosidase, which degrades cyanogenic glycosides. These pieces of evidence show the potential of fermentation to deal with glycosylation while protecting nutrients from thermal effects.

Our study highlights that culture is a fundamental driver of nutritional and pharmacological outcomes related to edible resources since it determines which parts of plants people consume and how they prepare them. Therefore, ignoring cultural variables in the analysis of antioxidant activity will produce inaccurate or wrong scientific conclusions.

This review connected data from different fields of knowledge. By analyzing the production of these areas in perspective, we highlight two main points that can inform future research focused on addressing broad problems related to food biodiversity. First, there is a gap concerning the nutritional and antioxidant potential of plants consumed by human populations. For example, ethnobiology studies report that people consume *D. grandiflora* seeds as food, but there is no research on their chemical profile and antioxidant potential. We suggest that in deciding what food resources to analyze, research teams prioritize plants already consumed by human populations, indicated by scientific studies (i.e., ethnobotany, ethnonutrition). Furthermore, food anthropology studies could prospect processing techniques used in Brazil to prepare legume plants. Second, there are some problems in data gathering and analysis in the main fields of knowledge we considered in this review. On the one hand, the food science community fails in providing the proper identification of plants. Plus, there is a lack of standardization in reporting antioxidant results, which makes interpretation and comparison between studies difficult. On the other hand, ethnobiological studies fail in presenting details on food processing. Papers from this community include two main types of errors: (1) mismatch between plant part and processing technique, (2) register of the product (e.g., tea) instead of processing technique (e.g., infusion). To overcome these main problems, it could be helpful to enroll multiple professionals while developing research protocols. More diverse teams certainly can design protocols capable of answering broader research questions that connect nutrition, local culture, and biological conservation, while avoiding basic problems in data gathering and analysis.

## Supporting information

**S1 Table. PRISMA checklist.**
(DOCX)

**S2 Table. List of studies reviewed.**
(DOCX)

## Acknowledgments

We thank teacher Jonathan who reviewed our English and polished our writing.

## Author Contributions

**Conceptualization:** Michelle Cristine Medeiros Jacob, Juliana Kelly da Silva-Maia, Fillipe de Oliveira Pereira.

**Data curation:** Michelle Cristine Medeiros Jacob, Juliana Kelly da Silva-Maia, Fillipe de Oliveira Pereira.

**Formal analysis:** Michelle Cristine Medeiros Jacob, Juliana Kelly da Silva-Maia, Fillipe de Oliveira Pereira.

**Funding acquisition:** Ulysses Paulino Albuquerque.

**Investigation:** Michelle Cristine Medeiros Jacob, Juliana Kelly da Silva-Maia, Fillipe de Oliveira Pereira.

**Methodology:** Michelle Cristine Medeiros Jacob, Juliana Kelly da Silva-Maia, Fillipe de Oliveira Pereira.

**Project administration:** Michelle Cristine Medeiros Jacob, Juliana Kelly da Silva-Maia, Fillipe de Oliveira Pereira.

**Resources:** Michelle Cristine Medeiros Jacob, Juliana Kelly da Silva-Maia, Fillipe de Oliveira Pereira.

**Software:** Michelle Cristine Medeiros Jacob, Juliana Kelly da Silva-Maia, Fillipe de Oliveira Pereira.

**Supervision:** Michelle Cristine Medeiros Jacob, Juliana Kelly da Silva-Maia, Ulysses Paulino Albuquerque, Fillipe de Oliveira Pereira.

**Validation:** Michelle Cristine Medeiros Jacob, Juliana Kelly da Silva-Maia, Ulysses Paulino Albuquerque, Fillipe de Oliveira Pereira.

**Visualization:** Michelle Cristine Medeiros Jacob, Juliana Kelly da Silva-Maia, Ulysses Paulino Albuquerque, Fillipe de Oliveira Pereira.

**Writing – original draft:** Michelle Cristine Medeiros Jacob, Juliana Kelly da Silva-Maia, Ulysses Paulino Albuquerque, Fillipe de Oliveira Pereira.

**Writing – review & editing:** Michelle Cristine Medeiros Jacob, Juliana Kelly da Silva-Maia, Ulysses Paulino Albuquerque, Fillipe de Oliveira Pereira.

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
