## [Decision Letter · Decision Letter 0]

22 Dec 2021

PONE-D-21-23830Culture Matters: A Systematic Review of Antioxidant Potential of Tree Legumes in the Semiarid Region of Brazil and Local Processing Techniques as a Driver of BioaccessibilityPLOS ONE

Dear Dr. Jacob,

Thank you for submitting your manuscript to PLOS ONE. After careful consideration, we feel that it has merit but does not fully meet PLOS ONE’s publication criteria as it currently stands. Therefore, we invite you to submit a revised version of the manuscript that addresses the points raised during the review process.

ACADEMIC EDITOR:Thank you for submitting your revised manuscript to PLOS ONE.  We believe it has merit but does not entirely fit PLOS ONE's publication standards as they currently stand and as suggested by Reviewer #1. As such, we invite you to submit an updated version of the manuscript as soon as possible that addresses the points mentioned by Reviewer #1 in order for us to make a decision on your article.  Additionally, we request that you make appropriate changes to the manuscript's formatting and reference section to ensure compliance with the journal's guidelines (please also refer to the Instructions for authors) Please ensure that your decision is justified on PLOS ONE’s publication criteria and not, for example, on novelty or perceived impact.

We look forward to receiving your revised manuscript.

Kind regards,

P. Balaji, Ph D

Academic Editor

PLOS ONE

Journal Requirements:

“We thank teacher Jonathan who reviewed our English and polished our writing. We also thank the National Institutes of Science and Technology in Ethnobiology, Bioprospecting, and Nature Conservation, certified by CNPq, with financial support from Facepe, the Foundation for Support to Science and Technology of the State of Pernambuco (Grant number: APQ-0562-2.01/17). This funding source had no role in the design of this study, nor in its execution, analyses, and interpretation of the data, and no role in our decision to submit results.”

“The National Institutes of Science and Technology in Ethnobiology, Bioprospecting, and Nature Conservation, certified by CNPq, provided financial support from Facepe, the Foundation for Support to Science and Technology of the State of Pernambuco to UPA (Grant number: APQ-0562-2.01/17). The funders had no role in study design, data collection and analysis, decision to publish, or preparation of the manuscript.”

Reviewers' comments:

Reviewer's Responses to Questions

**Comments to the Author**

1. Is the manuscript technically sound, and do the data support the conclusions?

Reviewer #1: Yes

Reviewer #2: Yes

2. Has the statistical analysis been performed appropriately and rigorously? 

Reviewer #1: Yes

Reviewer #2: Yes

3. Have the authors made all data underlying the findings in their manuscript fully available?

Reviewer #1: Yes

Reviewer #2: Yes

4. Is the manuscript presented in an intelligible fashion and written in standard English?

Reviewer #1: Yes

Reviewer #2: Yes

5. Review Comments to the Author

Reviewer #1: I have carefully read the Ms entitled {Culture Matters: A Systematic Review of Antioxidant Potential of Tree Legumes in the Semiarid Region of Brazil and Local Processing Techniques as a Driver of Bio-accessibility}, the Ms is very interesting, well and enjoyable written, well designed, easy to follow and informative. The authors have done great efforts to extract such interesting data and recommendations especially the conditions changing the acceptability of phenolic compounds and its cultural relationship.

I have some minor corrections

P2 correct “ (e.g( consisdine et al., 2017) ”

P2 correct “ (Ferreira-Júnior, W; Campos, L; Pieroni, A; Albuquerque, 2018) ”

P2L13, delete “see”

P3 correct “ (Ghoshal, 2018))”

Page 4 delete “see”

P5l3 rephrase “we synthesize the known ……..etc”

P6 no need to mention the names of the authors inside the experimental description (Data extraction section), it might be confusing to the respective reader; please rephrase this section

P8 L13-16, please add these data into table footnote

P49 correct (2013)found

Carefully check the format of scientific names inside the reference section, it should be italic ( e.g

Please correct Fig 2 legend (Page 8) and add symbols on the photos (a-i) not to describe as legend read from top-down as it might be confusing for some reader

Reviewer #2: Revised format of the paper Culture Matters: A Systematic Review of Antioxidant Potential of Tree Legumes in the Semiarid Region of Brazil and Local Processing Techniques as a Driver of Bioaccessibility is in an acceptable form as it meets the standards of the journal furthermore the language of the paper fulfils the requirement of the journal. The paper in its present format may be accepted for publication.

6. PLOS authors have the option to publish the peer review history of their article (what does this mean?). If published, this will include your full peer review and any attached files.

Reviewer #1: No

Reviewer #2: **Yes: **Dr C Pothiraj,

Assistant Professor,

PG Department of Botany,

Government Arts College,

Melur 625106 Madurai Tamil Nadu

India

---

## [Author Response · Author response to Decision Letter 0]

4 Jan 2022

Dear Editor,

We appreciate the careful reading of our manuscript. We have prepared a revised version of the paper that takes into account all of the comments made by the associate editor and reviewers. In the "Response to reviewer" file, we reply to the comments, specifying the text changes. 

We look forward to see our paper published.

Sincerely, 

Michelle Jacob

(on behalf of the authors)

---

## [Decision Letter · Decision Letter 1]

22 Feb 2022

Culture Matters: A Systematic Review of Antioxidant Potential of Tree Legumes in the Semiarid Region of Brazil and Local Processing Techniques as a Driver of Bioaccessibility

PONE-D-21-23830R1

Dear Dr. Jacob,

We’re pleased to inform you that your manuscript has been judged scientifically suitable for publication and will be formally accepted for publication once it meets all outstanding technical requirements.

Kind regards,

P. Balaji, Ph D

Academic Editor

PLOS ONE

Additional Editor Comments (optional):

Reviewers' comments:

Reviewer's Responses to Questions

**Comments to the Author**

1. If the authors have adequately addressed your comments raised in a previous round of review and you feel that this manuscript is now acceptable for publication, you may indicate that here to bypass the “Comments to the Author” section, enter your conflict of interest statement in the “Confidential to Editor” section, and submit your "Accept" recommendation.

Reviewer #2: All comments have been addressed

2. Is the manuscript technically sound, and do the data support the conclusions?

Reviewer #2: Yes

3. Has the statistical analysis been performed appropriately and rigorously? 

Reviewer #2: Yes

4. Have the authors made all data underlying the findings in their manuscript fully available?

Reviewer #2: Yes

5. Is the manuscript presented in an intelligible fashion and written in standard English?

Reviewer #2: Yes

6. Review Comments to the Author

Reviewer #2: The paper is good thoroughly revised have a work needs to be improved for its language to suit the standard of the journal.

7. PLOS authors have the option to publish the peer review history of their article (what does this mean?). If published, this will include your full peer review and any attached files.

Reviewer #2: **Yes: **Pothiraj Chinnathambi

---

## [Editor Report · Acceptance letter]

28 Feb 2022

PONE-D-21-23830R1 

Culture Matters: A Systematic Review of Antioxidant Potential of Tree Legumes in the Semiarid Region of Brazil and Local Processing Techniques as a Driver of Bioaccessibility 

Dear Dr. Jacob:

I'm pleased to inform you that your manuscript has been deemed suitable for publication in PLOS ONE. Congratulations! Your manuscript is now with our production department. 

Kind regards, 

on behalf of

Dr. P. Balaji 

Academic Editor

PLOS ONE